# Hemorrhoids Embolization: State of the Art and Future Directions

**DOI:** 10.3390/jcm10163537

**Published:** 2021-08-12

**Authors:** Alberto Rebonato, Daniele Maiettini, Alberto Patriti, Francesco Giurazza, Marcello Andrea Tipaldi, Filippo Piacentino, Federico Fontana, Antonio Basile, Massimo Venturini

**Affiliations:** 1Department of Radiology, Azienda Ospedaliera Marche Nord, Ospedale San Salvatore, Piazzale Cinelli 1, 61121 Pesaro, Italy; 2Division of Interventional Radiology, European Institute of Oncology, IRCCS, 20145 Milan, Italy; danielemaiettini@gmail.com; 3Department of Surgery, Azienda Ospedaliera Marche Nord, Ospedale San Salvatore, Piazzale Cinelli 1, 61121 Pesaro, Italy; alberto.patriti@ospedalimarchenord.it; 4Department of Vascular and Interventional Radiology, Cardarelli Hospital, Via Cardarelli 9, 80131 Naples, Italy; francescogiurazza@hotmail.it; 5Department of Surgical and Medical Sciences and Translational Medicine, Sapienza-University of Rome, 00189 Rome, Italy; tipaldi.andrea@gmail.com; 6Diagnostic and Interventional Radiology Department, Circolo Hospital, ASST-Sette Laghi, 21100 Varese, Italy; filippo.piacentino@asst-settelaghi.it (F.P.); federico.fontana@uninsubria.it (F.F.); massimo.venturini@uninsubria.it (M.V.); 7Department of Medicine and Surgery, Insubria University, 21100 Varese, Italy; 8Radiodiagnostic and Radiotherapy Unit, Department of Medical and Surgical Sciences and Advanced Technologies, University Hospital “Policlinico Vittorio Emanuele”, 95123 Catania, Italy; basile.antonello73@gmail.com

**Keywords:** hemorrhoids, embolization, interventional radiology, bleeding, coil embolization

## Abstract

Hemorrhoidal disease is a frustrating problem that has a relevant impact on patients’ psychological, social, and physical well-being. Recently, endovascular embolization of hemorrhoids has emerged as a promising mini-invasive solution with respect to surgical treatment. The purpose of this article is to review the indications, technical aspects, clinical outcomes, and future prospective of endovascular embolization of symptomatic hemorrhoid patients.

## 1. Introduction

Hemorrhoid disease (HD) is the most common anorectal condition; epidemiological data are difficult to collect and range widely between 4 and 35% in the literature, depending on several variables (such as the county of the study and methodology of assessment) [1,2,3,4]. Hemorrhoids are the manifestation of vascular enlargement of anal cushions due to several factors [5]. Most patients are relieved by medical or office-based (injection sclerotherapy, rubber band ligation, infrared photocoagulation, bipolar diathermy) treatments, and surgical treatment is necessary in 10% of cases. The reference surgical procedures are Milligan and Morgan’s open hemorrhoidectomy and Ferguson’s closed hemorrhoidectomy. However, nowadays less invasive techniques are available such as circular stapled anopexy (Longo procedure) and elective Doppler-guided hemorrhoidal artery ligation (DG-HAL) [5]. By analogy, comparable arterial occlusion was proposed with embolization: the “emborrhoid” technique was recently introduced by Vidal et al. [6,7].

## 2. Etiology

To understand hemorrhoidal disease, we should always consider the physiology of the rectum and hemorrhoidal vascularization. The major rule of the hemorrhoids is to participate in anal continence; Lestar et al. proved that the contribution of the hemorrhoidal plexuses to the maximal anal basal pressure is about the 20% [7]. The functional unit of hemorrhoids is the *corpus cavernosum recti* (CCR). The distal branches of superior rectal arteries (SRA) enter the muscularis mucosae of the rectum at the level of S3, and some of the terminal branches directly run in the corpus, acting as the filler of the CCR (usually hemorrhoidal bleeding is bright red) [8]. The corpus presents dilated hollow spaces communicating with each other in a manner resembling “a net”, and they are separated from each other by tough connective tissue septa, rare muscular cells, presenting a glomerular appearance. Veins lie between these spaces and anastomose with each other freely, acting as the drainage of the CCR [4]. These veins represent the venous hemorrhoidal plexus and drain both in the portal and systemic system; so, the CCR has no direct connection to the portal or cava vein [9]. In 2006, Aigner et al. presented their data demonstrating that arterial hyperplasia both of the submucosal rectal arteries and transmural branches of the SRA correlates significantly with the appearance and the grade of HD [10]. The comprehension of the arterial component of the HD and the positive results of the Doppler-guided hemorrhoidal artery ligation (DG-HAL) [11] led to the development of the “emborrhoid technique” [6].

The causes of HD development are still not well known and seem to be multifactorial including several patient-related variables such as toileting behavior and diet. HD patients often have incorrect toilet behaviors such as spending a long time on the toilet, repeated straining, or using their phone while on the bathroom. A poor fiber diet demonstrated to correlate with HD as well as inadequate liquid intake [12,13].

## 3. Clinical Presentation

Several benign anorectal conditions present with overlapping symptoms. Both unfamiliar physicians with anorectal conditions and patients often address all anorectal symptoms to hemorrhoids [3]. The most common and known symptom of internal hemorrhoids is painless bright red bleeding with stools, on the toilet paper, dripping, or even squirting into the toilet water. Other symptoms are burning, pain or itching, or feeling pressure. Since the first classification by Goligher in 1975, a different classification has been proposed for HD, usually divided into four grades characterized by the principal presentation of prolapse and bleeding and from possible additional features such as pain, pruritus, discomfort, and skin tags [14,15].

## 4. Conservative Management

In case of low-grade hemorrhoids, the first line treatment is to modify the patient’s behavior associated to diet support. A high-fiber diet and adequate liquid intake during the day are the first step in conservative management. Diet should be mandatory associated to defecation behavior modification such as avoiding straining, avoiding spending a long time sitting on the toilet reading or using a smartphone (a habit that is more and more widespread nowadays) [12,13,16]. Laxatives could be introduced as an aid to the diet to soften feces. Topical and oral medications can be used when needed to control symptoms; usually, analgesics or phlebotonic agents (e.g., flavonoids) are used [17]. Regardless of the surgical, endoscopic, or endovascular treatments used, a lifestyle modification with adequate daily hydration and medical therapy are essential to obtain clinical success [16].

## 5. Arterial Anatomy

The hollow spaces of the CCR are feed directly into the thick-walled artery without interposed capillaries. The blood remains arterial and acts as a “filler”, playing no metabolic role similarly to *corpus cavernosum penis* [4]. For this reason, bleeding from hemorrhoids is always bright red. The CCR represents a direct arteriovenous communication between the terminal branches of rectal arteries and their corresponding veins. The rectum is supplied by three major arteries: the superior rectal artery (SRA), the middle rectal wall artery (MRA), and the inferior rectal artery (IRA).

The most prevalent arterial supply of the CCR derives from SRA and, therefore, from the inferior mesenteric artery (IMA). The IMA comes off the anterolateral aspect of the left side of the aorta just above the bifurcation of the common iliac arteries usually at L3. The branches of the inferior mesenteric artery include: (1) the left colic, which is located retroperitoneally and supplies the descending colon; (2) the sigmoid branches, which supply the sigmoid colon; (3) the superior rectal artery, which is the terminal branch of the inferior mesenteric artery at the pelvic rim. The main trunk of the superior rectal artery bifurcates, in the center, about 12 cm above the dentate line, in four (occasionally three or five) large branches, two posterior (one right and one left) and two lateral branches (one right and one left); each branch partially spiraled around the central axis of the rectum and subsequently separated into 5–7 branches penetrating the bowel wall 3–4 cm above the elevators ani and traverses the muscle wall. About 2–3 cm above the dentate line, twisting arteries of maximum diameter 2 mm emerge towards the mucosal surface [9].

In contrast, the IRA supplies blood to the anal canal and anus. Its branches from the internal pudendal artery passes in the ischiorectal fossa through the external anal sphincter and, finally, runs into the wall of the rectum. The IRA is present on both sides in almost all individuals. The SRA and IRA are very stable anatomical structures that are found in almost all individuals [9]. Although the SRA and IRA communicate with each other, the connection is primarily observed within the submucosal layer of the rectum, rather than outside the wall. The described “orthodox” anatomy of SRA is found in the 46% of patients and was described by Thomson as type I (Figure 1) [9]. The other 54% of cases present high variability that was reassumed in four types:Type II: the main trunk passes to the right, giving off branches to the left.Type III: the main trunk passes to the left supplying branches to the right and a MRA define a relevant contribution to the vascularization of the CCR.Type IV: trifurcation of the main trunk with supplying branches to both sides of the anal canal in some with a minor participation of MRA to the vascularization of the CCR (Figure 2).Type V: trifurcation of the main trunk with supplying branches to both sides of the rectum not reaching the anal canal with an MRA defining a main contribution to the supply of CCR.

The frequency of the middle rectal artery is reported to range from 12 to 97% and presents several variances in its origin and path [18]. The variability of MRA origin is well known from the experience obtained with prostate arterial embolization (PAE). Didio reported an origin of MRA from internal pudendal artery (IPA) and inferior gluteal artery (IGA) in 40% and 26%, respectively [19]. The knowledge of the anatomy of the MRA is fundamental for the understanding and management of patients treated by endovascular route. Thomson found in 10% of cases the type V anal vascularization that defined “interesting and previously undescribed vascular anomaly”: in this pattern, the MRA supply most of the anal canal vascularization usurping the function of the superior rectal artery. However, the anatomy studies available nowadays are obtained on the cadavers of healthy subjects not affected by hemorrhoidal pathology. In consideration of the prominent role of vascular abnormalities at the basis of hemorrhoidal pathogenesis (arterial hyperflow) [16], it is likely that in subjects affected by hemorrhoids the vascular pattern is different from that reported in healthy subjects. In the results of a prospective clinical study on the efficacy and safety of superior rectal artery embolization of hemorrhoidal disease as a first-line invasive treatment, Tradi reported an incidence of the type V pattern of vascularization in 24% of hemorrhoid patients. The relevance of the fundamental role of MRA in the CCR perfusion is supported by the fact that in most of the studies available, despite a primary technical success of complete SRA occlusion below the pubic symphysis, the clinical relapses are often supported by an angiographic finding of prominent MRAs supplying the corpus cavernosum of the rectum [20,21,22]. Hence, the rationale for the use of particles during the embolization of the SRA so as to occlude the SRA-MRA connections and the hypothesis sustained by several authors to occlude the MRA whenever visible from the angiographic examination of the SRA.

## 6. Embolization Agents

Among several embolization agents available nowadays in interventional radiology practice, to our knowledge, until now, only two have been reported in the emborrhoid technique: coils and particles.

Coils were first described by Vidal et al. for the emborrhoid technique and were considered by authors for the low risk of bowel ischemia defined by the proximal occlusion avoiding distal embolization [20]. Usually, 2 or 3 mm diameter coils are used to occlude all the branches of the SRA above the pubic symphysis. Giurazza et al. proposed in 2020 the “spaghetti technique”, using oversized coils released in a stretched fashion, in patients with portal hypertension and chronic hemorrhoidal bleeding. This technique helps to avoid a sudden occlusion of the vessel and to use a smaller number of coils [23]. Since the anastomosis between SRA/MRA could be located distally, metallic coils could not guarantee a correct devascularization. In 2016, the first study was published reporting the combined use of polyvinyl alcohol (PVA) particles and coils [21]. The rational for the application of particles is to achieve a more distal embolization that on one hand could increase the risk of rectum ischemia, but on the other hand, will provide a distal embolization at the level of hemorrhoidal plexus. The 300 µm PVA particles supposed to provide an ischemia closer to the hemorrhoidal plexus and, probably more important, could provide the embolization of the anastomotic branches with middle and inferior rectal arteries. The use of particles also solves the issue of the recanalization of coils. Four papers reported the use of tris-acryl gelatin particles (TAGp); no comment on the rationale to TAGp use versus PVA particles was done by the authors [22,24,25,26]. Nowadays, no data on the caliber of the anastomosis are available to understand the right particle size. Küçükay et al. performed a randomized trial to understand the best tris-acryl gelatin particles size (500–700, 700–900, and 900–1200 μm) [25]. The earliest significant bleeding control was achieved with 500–700 μm particles, and the best bleeding control and lower complication rate was achieved with 900–1200 μm particles at 12 months [24,25]. A recent retrospective analysis to compare safety and clinical outcomes of embolization using particles and coils versus coils only reported a technical success of 100% and no major complications in both groups. Clinical success was obtained in 66% in patients with no difference when using combined embolization with particles and coils versus coils alone [26].

To solve the distal embolization issue, the hypothesis to use liquid embolic agents such as N-butyl-cyanoacrylate (NBCA) and ethylene vinyl alcohol copolymer (EVOH) was recently introduced [27]. The author supposed that the successful results achieved with NBCA and EVOH in neurological and peripheral embolization could be replicated in hemorrhoids treatment, ensuring a distal penetration in small anastomosis, eliminating possible connections and revascularization of the hemorrhoidal plexus from other supply branches. Especially, EVOH, in addition to ensuring elastic, non-absorbable, non-adhesive, and permanent liquid embolization with a characteristic progressive solidification, also provides high vascular penetration with minimal inflammatory effects on the endothelium.

Although speculation on liquid agents seems to have good logic, case series are not yet available in the literature, but only the case reported by Alonso-Burgos in his commentary. For emborrhoid, the cost can vary in different countries. The estimated cost ranges from EUR 800 to 2000 depending on the type and number of embolic agents and microcatheter used.

## 7. Clinical Indications

The indication for the treatment of hemorrhoids reported in the literature is variable, with studies suggesting to treat only bleeding hemorrhoids grades I–III other than generally symptomatic grade I–III hemorrhoids (therefore, considering the associated symptoms such as burning, pain, itching) not responsive to conservative management [6,20,21,22,26,27,28].

It is the opinion of the authors to consider bleeding hemorrhoids as the main indication of emborrhoid, as it represents the most appropriate justification for intra-arterial occlusion so to avoid any criticism in case of clinical failure or ischemic damage to the anal canal.

The emborrhoid technique does not allow us to treat grade IV hemorrhoids with non-reductive prolapses nor does it allow us to perform mucopexy lifting, as in the case of THD-HAL with benefits on the associated painful and irritative symptoms.

## 8. Hemorrhoid’s Embolization Outcome and Results

The data available from the literature are still too few to be able to define validated conclusions on scientific evidence. As reported in the Table 1, only seven publications refer to more than 10 patients for a total amount of less than 230 cases. Papers reported different follow-up periods ranging from 1 to 12 months, with different and clinical scores for outcome that hamper a scientifically valid comparison. Most of the clinical studies come from French Departments [6,20,22,26,27].

All studies defined technical success rate as the ability to close all branches of SRA below the pubic symphysis with results that range between 90% and 100%. The high technical success described by various authors could confirm the low complexity of this procedure, which is very import to allow the spread of the technique in new centers that want to approach this new intervention for HD treatment (Figure 1 and Figure 2). Compared to PAE, a successful embolic procedure recently emerged in clinical practice, emborrhoid is technically simpler and with shorter execution times (on average 1 h) and less exposure to ionizing radiation.

Additionally, safety of the procedure represents an important issue when introducing a new technique. All articles reported no major complication. Coil embolization is generally considered embolizing bowels, while introducing particles raised concerns, particularly for ischemic complications. When using PVA, no anal pain syndrome was reported [21]; Zakharchenko et al. performed rectoscopic examinations at one-day, one-week, and one-month intervals after the procedure and found no mucosal atrophy or postembolization dystrophy with histopathological analysis.

Abd El Tawab et al. described the first experience with TAGp (300–500 μm) reporting postembolization anal pain 27 of 33 (81%), but the symptoms were self-limiting and lasted a few days; no other complications were reported [24]. Küçükay et al. reported a 100% rate of post procedural pain using TAGp 500–700 μm, 64% using 700–900 μm, and no pain using 900–1200 μm. Moreover, using particles less than 900 μm showed minor complication: 45% rate of superficial ulcerations (<5 mm) at the embolized area, 7% of small recto-sigmoid junction ulcerations, and 2% of small fibrotic scar tissue in the treated zone. The only comparative study between different sizes of TAGp showed similar results in terms of HD control in all particle dimensions groups [25].

Regarding the clinical outcome, the published studies adopted different evaluation scales, hampering a comparison. Clinical success ranges between 63 and 97%. Defining the period of clinical success is also critical because authors reported different observation periods, and as above discussed, medical treatment and behavior modification are essential to ensure the long-lasting effect of treatments. The absence of a standardized medical treatment and symptoms reporting scale together with behavior variability make the results difficult to compare.

Coil embolization provides a proximal embolization of SRA; in such cases, distal anastomosis feeding the CCR can be activated and cause relapse. Moussa et al. demonstrated that adding MRA embolization to SRA embolization with particles (embospheres 300–500 μm) and coils showed improved clinical symptoms without any major complications in two patients who had clinical failure after SRA embolization with coil embolization [26].

Even if the data available are still too few, the endovascular embolization compared with the other surgical treatment, avoiding any anorectal manipulation, has the advantage to treat those patients that for systemic comorbidities (i.e., respiratory or cardiovascular compromission, anticoagulation, or antiplatelet therapies) or local pathologies (i.e., fecal incontinence, inflammatory bowel disease, proctitis, and previous radiotherapy) cannot be approached with classic or mini-invasive surgical interventions [21].

## 9. Futures Perspectives

The emborrhoid technique is a simple, new, and minimally invasive procedure available in clinical practice dealing with HD [28,29,30,31]. HD is a frustrating problem that could impact on a psychological, social, and physical points of a patient’s wellness.

Even if the preliminary study available and discussed in this paper gives encouraging results in term of efficacy and safety, further studies are mandatory to deeply understand the right application of such a technique. Probably the role of MRA-SRA connection is the key feature for good clinical results, and the concept of occluding the MRA whenever visible in the selective angiography of the SRA could be the correct message for the interventional radiologist approaching the hemorrhoidal pathology.

Anatomical studies to better understand CCR vascular components and the caliber of the anastomoses between SRA-MRA-IRA in order to understand the best embolic agent could be useful.

Prospective studies with large case series comparing the different embolic agents and medium- and long-term follow up are necessary to better define the efficacy and safety of the procedure. Comparative studies with established and scientifically validated techniques such as DG-HAL or the Longo technique are equally desirable.

## Figures and Tables

**Figure 1 jcm-10-03537-f001:**
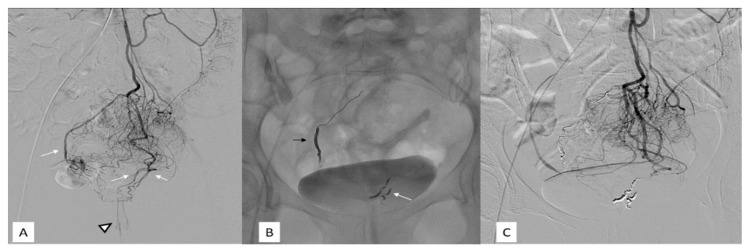
Fifty-two-year-old female with grade IV hemorrhoids disease and a low quality life and inability to do a regular work. (**A**) Subtracted digital arteriography of the inferior mesenteric artery. Visualization of the rectal branches (white arrows) and of the CCR arteries (black arrowhead) with a type 1 vascularization according to the Thomson classification. (**B**) Non-subtracted single-shot fluoroscopy after embolization with metallic coils of the sigmoidal arteries. On the left, a proximal embolization with a packing coil (black arrow). On the right, a more distal embolization with regular coils (white arrow). (**C**) Subtracted digital arteriography of the inferior mesenteric artery after embolization shows complete exclusion of the CCR arteries (no vascularization below the pubic symphysis).

**Figure 2 jcm-10-03537-f002:**
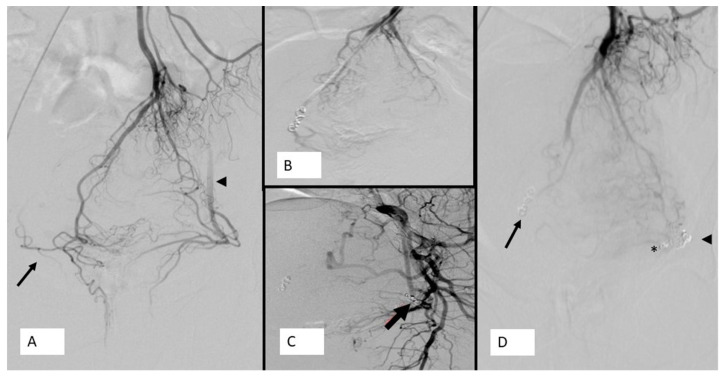
Forty-three-year-old lady, weight lifter, suffering from grade III bleeding hemorrhoids: (**A**) Angiogram of SRA showing a type IV vascularization according to the Thomson classification After the coiling of left lateral branch of SRA (arrow), a selective angiogram of left internal iliac artery showing a patent left MRA. (**B**) Coiling of right lateral branch of SRA (arrow). (**C**) of the CCR feeding artery below the pubic symphysis. (**D**) The final angiogram of SRA showing coiling of right lateral branch of SRA (arrow), coiling of left MRA (asterisk), and coiling of left lateral branch of SRA (arrowhead), with occlusion with evidence of retrograde filling of right MRA (arrow) and left MRA (arrowhead). Note the diameter of left MRA is equal to that of the SRA.

**Table 1 jcm-10-03537-t001:** Clinical studies 2014–2021.

Study	Pt. Number	Country	Patient Selection	Embolic Agent	Median FU	Complication	Outcome (% of No/Unsatisfactory Clinical Success)
Vidal 2014	14		Goligher’s Grade II or IV	Coils	6 months	none	28%
Zakharchenko 2016	40	France	Goligher’s Grade I to III	Coils + PVA	1 month	none	6–17%
Moussa 2016	30	France	Goligher’s Grade I to IV	Coils	5 months	none	28%
Tradi 2018	25	France	Goligher’s Grade II or III	Coils	12 months	none	28%
Giurazza 2020	5	Italy	portal hypertension and chronic hemorrhoidal bleeding	Colis	3 months	none	20%
Abd El Tawab 2020	33	Egypt	Goligher’s Grade II or III	TAGp	12 months	none	3%
Moussa 2020	38	France	Goligher’s Grade II or III	Coils + TAGp	6 months	15.00%	34.00%
Küçükay 2021	42	Turkey	Goligher’s Grade I or IV	TAGp	12 months	54%	7%

PVAp: polyvinyl alcohol particles; TAGp: tris-acryl gelatin particles.

## Data Availability

All articles used in this review are available on PubMed.

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
