# Peer review of "Hemorrhoids Embolization: State of the Art and Future Directions"

_jcm, 2021, doi:10.3390/jcm10163537_

Round 1
Reviewer 1 Report
It seems to me an interesting paper that makes an exhaustive review of the hemorrhoidal embolization technique, with the new variants of the original technique (particles + coils). However there are some writing errors to review (for example line 37...).
Finally, in line 212, there is a statement in which I do not agree at all, I do not think that the "emborrhoid" technique presents a "enormous" benefit over the THD-HAL technique due to the fact of identifying the arterial branches, since that the use of Doppler allows this identification at the distal level. I believe that the main benefit of the emborrhoid technique lies in the absence of anorectal manipulation in those patients with different pathologies in which a surgical hemorrhoidectomy can generate fecal incontinence or inflammatory problems (those patients with a history of fecal incontinence or Inflammatory Bowel Disease with proctitis). Furthermore, the use of a technique such as THD-Hal could present some benefits over the emborrhoid technique in healthy patients, since it allows the performance of hemorrhoidal mucopexy that could improve hemorrhoidal prolapse, unlike embolization, which would have little impact on this aspect of symptomatology.
In summary, it seems like a good paper, but it does not make the indications of the technique in a clear way, nor when in which cases it would be preferable to a surgical hemorrhoidectomy or THD-HAL.
Author Response
I want to thank the reviewers for the comments and suggestions.
Reviewer 1
-line 212, according to the reviewer’s suggestion we would change the paragraph lines 211-213 with the following: “Even if the data available are still few the endovascular embolization compared with the other surgical treatment , avoiding any anorectal manipulation, has the advantage to treat those patients that for systemic comorbidities ( i.e. respiratory or cardiovascular compromission , anticoagulation or antiplatelet therapies ) or local pathologies ( i.e. fecal incontinence , Inflammatory Bowel Disease, proctitis, previous radiotherapy ) cannot be approached with classic or mini-invasive surgical interventions [21].
-In the Clinical Indication we would add the following phrase: “the embrorrhoid technique does not allow to treat grade IV hemorrhoids with non-reductive prolapses nor does it allow to perform mucopexy lifting , as in the case of THD-HAL with benefits on the associated painful and irritative symptoms”.
Reviewer 2 Report
- It should be mentioned, that all cited publications come from only 3 departments.
- It should be stated with more caution, that the follow up is rather short, no differentiation of the hemorrhoidal grades, very small sample size, no details of postoperative resolution of symptoms.
- It also should be mentioned, that complications might low but life threatening (pelvic sepsis if necrosis of the bowel wall might happen).
- There should be one section added that discuss costs of the procedure
- In the future section the recommendation should be drawn with much more caution.
Author Response
I want to thank the reviewers for the comments and suggestions.
Point to point reply reviewer 2 :
Reviewer :It should be mentioned, that all cited publications come from only 3 departments.
Authors:The bibliography used consists of articles on haemorrhoidal pathology published in journals of human anatomy, gastroenterology, surgery and interventional radiology with a time frame ranging from 1964 to 2021.
Reviewer: It should be stated with more caution, that the follow up is rather short, no differentiation of the hemorrhoidal grades, very small sample size, no details of postoperative resolution of symptoms.
Authors: according to the reviewer’s suggestions in 8.Hemorrhoid’s embolization outcome and results section we would modify the lines 180-182 with the following paragraphs: “ the data available from the literature are still few to be able to define validated conclusions on scientific evidence. As reported in the table 1 only 7 publications refer to more than 10 patients for a total amount of less than 230 cases . Papers reported different follow-up periods ranging from 1 to 12 months , with different and clinical scores for outcome that hamper a a scientifically valid comparison. “
Reviewer: It also should be mentioned, that complications might low but life threatening (pelvic sepsis if necrosis of the bowel wall might happen).
Authors: in the literature to date and to our knowledge no procedure-related deaths or major morbidities as pelvic abscesses or intestinal infarcts have been reported. In the recent and interesting Turkish study pubblished in JVIR ( 25) the author evaluated the safety and efficacy of superior rectal artery embolization (SRAE) with different-sized tris-acryl gelatin microspheres by rectoscopy, clinical examination, and questionnaires with a follow up of 12 months. They reported only minor complication rate (n = 23/42) in the treated zone: 45% small superficial ulcerations (n = 19/42), 7% small rectosigmoid junction ulcerations (n = 3/42), and 2% small fibrotic scar tissue (n = 1/42).
Reviewer :There should be one section added that discuss costs of the procedure.
Authors: a cost assessment is not feasible due to the different material tariffs applied in the countries. However, both the coils and the particles have rather low costs compared to more expensive materials used in interventional radiology such as covered stents or non-adhesive liquids (Onyx). We would add the following phrase in the 6.Embolization Agents unit: “the embolic agents used up to now are relatively cheap with a mean cost ranging from 300 up to 1000 € according the different country”
Reviewer: In the future section the recommendation should be drawn with much more caution.
Authors: according to the reviewer’s suggestions we would modify the last sentence lines 227-228 “ Comparative studies ….. equally desirable “ with“ Comparative studies with established and scientifically validated techniques such as DG-HAL or the Longo technique are equally desirable”
Round 2
Reviewer 2 Report
Thanks for your reply. There seems to be a misunderstanding:
1. It should be mentioned, that all cited publications come from only 3 departments.
This is only focussed on Tab 1. not on the whole bibliography ... Therefore it should be included in one sentence.
2. There should be one section added that discuss costs of the procedure.
This means complete costs of the procedure (estimated) not only the material. (Why this: comparable treatments like rubber band ligations are much cheaper).
All the rest is sufficiently answered.
Author Response
Thank to the riviewer for the comments.
- It should be mentioned, that all cited publications come from only 3 departments.
We inserted a sentence in the text (line 188) emphasizing that most of the studies (6,20,22,26,27) come from three French departments. However, I would like to point out that many other studies from other countries are reported in the bibliography: the studies of Giurazza (23) and Venturuini (29) are Italian, that of Abd El Tawab (24) is Egyptian, that of Küçükay (25) from Turkey, that of Zakharchenko (21) from Russia.
2. There should be one section added that discuss costs of the procedure.
The sentence about the cost was modified .
Cordially
A Rebonato